# Targeting FGFR Pathways in Gastrointestinal Cancers: New Frontiers of Treatment

**DOI:** 10.3390/biomedicines11102650

**Published:** 2023-09-27

**Authors:** Margherita Ratti, Elena Orlandi, Jens Claus Hahne, Stefano Vecchia, Chiara Citterio, Elisa Anselmi, Ilaria Toscani, Michele Ghidini

**Affiliations:** 1Oncology and Hematology Department, Piacenza General Hospital, Via Taverna 49, 29121 Piacenza, Italy; 2Centre for Evolution and Cancer, The Institute of Cancer Research, London SM2 5NG, UK; 3Pharmacy Unit, Piacenza General Hospital, Via Taverna 49, 29121 Piacenza, Italy; 4Oncology Unit, Fondazione IRCCS Ca’ Granda Ospedale Maggiore Policlinico, 20122 Milan, Italy

**Keywords:** FGFR pathways, gastrointestinal cancers, target therapies

## Abstract

In carcinogenesis of the gastrointestinal (GI) tract, the deregulation of fibroblast growth factor receptor (FGFR) signaling plays a critical role. The aberrant activity of this pathway is described in approximately 10% of gastric cancers and its frequency increases in intrahepatic cholangiocarcinomas (iCCAs), with an estimated frequency of 10–16%. Several selective FGFR inhibitors have been developed in the last few years with promising results. For example, targeting the FGFR pathway is now a fundamental part of clinical practice when treating iCCA and many clinical trials are ongoing to test the safety and efficacy of anti-FGFR agents in gastric, colon and pancreatic cancer, with variable results. However, the response rates of anti-FGFR drugs are modest and resistances emerge rapidly, limiting their efficacy and causing disease progression. In this review, we aim to explore the landscape of anti-FGFR inhibitors in relation to GI cancer, with particular focus on selective FGFR inhibitors and drug combinations that may lead to overcoming resistance mechanisms and drug-induced toxicities.

## 1. Introduction

The fibroblast growth factor receptor (FGFR) family includes four transmembrane receptor tyrosine kinases: FGFR1, FGFR2, FGFR3 and FGFR4 [1], and FGFR5 or FGFRL1, a fifth family member lacking the tyrosine kinase domain [2] (Figure 1).

FGFRs are expressed on the cell membrane and consist of three main components: three extracellular Ig-like domains, a transmembrane helix and an intracellular tyrosine kinase domain. The native ligands of FGF receptors are FGFs, which belong to a family including 22 members, grouped into subfamilies; 18 encode molecules known to signal through FGF tyrosine kinase receptors. Receptor dimerization starts after binding to the ligand, which causes a conformational change in the receptor structure. This conformational change triggers an intracellular kinase domain transautophosphorylation event, which activates downstream signal transduction pathways [3] (Figure 1). The FGFR pathway is subject to various somatic aberrations resulting in carcinogenesis or therapy resistance across a wide range of tumor types, among them gastric cancer (GC), pancreatic cancer, colorectal cancer (CRC) and cholangiocarcinoma (CCA) (Figure 2) [4,5].

Deregulated FGF signaling in tumors can be mediated by receptor amplification, mutation and gene fusion (Table 1). 

In addition, ligand-dependent activation results in oncogenic aberrations and represent features that make FGFR an ideal therapeutic target. Gene amplification is the most common aberration of the FGFR2 gene amongst gastrointestinal cancers [16], with differences between early and advanced stages: amongst the early stages, FGFR2 amplifications are associated with larger tumor size (T stage) and major lymph node dissemination, which can lead to poorer patient outcomes, including inferior overall survival. In the metastatic setting, FGFR2 amplifications are associated with inferior progression-free survival and overall survival in patients receiving chemotherapy with platinum and fluoropyrimidine. In GC, FGFR2 and HER2 amplifications are mutually exclusive, although rare cases have been reported. FGFR2 amplification occurs in microsatellite stable tumors and is not related to PD-L1 expression. FGFR2 and FGFR3 have comparatively more frequent gene rearrangements [6]. FGFR expression in colon cancer is correlated with tumor progression [17]. Oncogenic signaling via the FGFR pathway has emerged as a targetable site in multiple cancers; therefore, assessment of the structure of FGFRs and FGFR inhibitors is being continuously developed.

Currently, drugs targeting the FGF/FGFR signaling pathway include tyrosine kinase inhibitors (TKIs), monoclonal antibodies and FGF ligand traps [18]. The first approach for FGFR targeting included oral TKIs; in fact, their structures are similar to adenosine triphosphate (ATP), competing for the ATP binding cleft of the kinase domain on the FGFR receptor. The reversible binding reduces tyrosine kinase phosphorylation, causing blockade of multiple downstream pathways, thereby inhibiting cancer cell proliferation [19]. Due to the increased toxicity, the FGFR pathway is involved in normal eye and nail development but also in phosphatemia homeostasis; common adverse effects are hyperphosphatemia gastrointestinal disorders, skin, nail and ocular disorders and inability to accurately predict response; a first-generation TKI was quickly sidelined. Development of irreversible FGFR inhibitors improved the duration of the activity and related toxicities are more easily managed with a low-phosphate diet, phosphate binders and supportive interventions [20]. Several selective TKIs were approved for clinical use, FGFR2 fusions were found in 10–15% of intrahepatic cholangiocarcinomas (iCCAs) and many TKIs showed promising results in phase II clinical trials [21]. Monoclonal antibodies target FGFRs and interfere with ligand binding and receptor dimerization, demonstrating fewer side effects due to an antibody-dependent cell toxicity [16]. Current treatment strategies are mostly inclined to combination therapy with chemotherapy or immunotherapy [22].

## 2. FGFR Pathway and Its Alterations in Gastrointestinal Cancers

The FGFR pathway is involved in many physiological conditions (e.g., embryogenesis, angiogenesis, wound healing, tissue homeostasis, cell differentiation, proliferation, migration and apoptosis) and aberrant activation can cause pathological conditions (e.g., chronic kidney disease and cancers) [3,23,24,25] (Figure 1). The FGFR pathway is composed of five highly homologous receptor tyrosine kinases (FGFR1-5) and eighteen secreted extracellular ligands (fibroblast growth factors, FGFs) which act in a paracrinal and endocrinal manner [18,24,26] FGFR1-4 are composed of three Ig-like extracellular domains, followed by a hydrophobic transmembrane helix and an intracellular tyrosine kinase domain. These four receptors exert tyrosine kinase activity [27,28]. In contrast, FGFR 5 (also known as FGFRL1) has no tyrosine kinase activity and acts as a decoy receptor involved in preventing excessive activation of this signaling pathway [2,29]. The receptor tissue distribution and substrate-binding selectivity of FGFRs are different for each receptor. Furthermore, the fact that the FGFR genes can be spliced into 48 distinct isoforms increases the heterogeneity and ligand specificity of the resulting isoforms [30]. FGFRs are activated by receptor homo- and heterodimerization caused by the binding of FGF to the receptors. After receptor dimerization, a conformational shift induces an intracellular transautophosphorylation, which in turn results in the activation of downstream signaling pathways [18]. FGFRs require the cytosolic FGFR-associated interacting protein FRS2 for signal transmission and activation of the PIK3-AKT-mTOR, RAS-RAF-MEK-ERK and JAK-STAT intracellular signaling pathways [31] (Figure 1). Furthermore, FGFRs can be activated either by overexpression of ligands or in a ligand-independent manner. Such deregulated FGFR signaling is often observed in several cancer entities and is involved in tumor growth and angiogenesis [32] (Figure 1).

The mechanisms of FGFR signaling deregulation (Table 1) can be caused by:-FGF overproduction

Autocrine FGF overproduction has often been described in tumors [3]. One example is FGF-5 overexpression in esophageal and colon cancer [33], as well as FGF-2, FGF-8, FGF-17, FGF-18 and FGF-19 overexpression in hepatocellular tumors and head and neck squamous cell carcinoma [34,35,36,37,38]. Furthermore, paracrinal effects, especially from FGF-2, have been described. Increased level of FGF-2 results in upregulated expression of anti-apoptotic proteins as well as in promotion of tumor neoangiogenesis caused by activation of FGFR-1 and -2, the main endothelial FGFRs [39].

-FGFR mutations

Some mutations resulting in constitutive activation of FGFRs are well known. Mutations in FGFRs are found in nearly 26% of cancers with FGFR gene abnormality [40]. Very often, activating mutations are located in the extracellular and kinase domains of the receptors FGFR-2 and FGFR-3 in human tumors [6].

-Chromosomal translocation

Chromosomal translocation resulting in gene fusion of the FGFR kinase domain with other constitutively expressed genes has been described in some gastrointestinal cancers [40,41]. The fusion proteins are permanently dimerized even in the absence of ligands, no longer under feedback inhibition control and, therefore, exerting permanent signaling [42,43]. Examples are FGFR2 fusion proteins in CCAs [44] and FGFR-1 fusion proteins in gastrointestinal stromal tumors [12].

-FGFR gene amplification

FGFR gene amplification results in FGFR overexpression leading to ligand-independent signaling. In 10% of all GC, an amplification of FGFR-2 is present, which is correlated to poor prognosis [45]. Sometimes, FGFR-2 amplification accompanies deletion of the C-terminal coding exon. This prevents receptor internalization and results in constitutive receptor activation [46].

-Altered FGFR splicing

Altered gene splicing of FGFRs results in upregulation of ligand-dependent signaling. Due to the altered splicing, FGFRs become targets for a broader range of FGFs [47]. One example is the altered splicing of the IgIII domain in FGFR-1, -2 and -3, which leads to a changed receptor-binding affinity towards FGFs found in the healthy stroma and creates an aberrant paracrine signaling loop [48].

-Germline single-nucleotide polymorphisms

Some single-nucleotide polymorphisms (SNPs) have been identified within the FGFR-2 and FGFR-4 genes. These SNPs cause increased receptor stability and they are most probably associated with an increased risk for, as well as a more aggressive form of, breast and colon cancer [49,50,51].

## 3. Targeting FGFR in Cholangiocarcinoma and Pancreatic Cancer

Alterations in FGFR2 were found in approximately 14% of patients with iCCA [52]. The selective FGFR inhibitors pemigatinib, infigratinib and futibatinib have been approved by the Food and Drug Administration for the treatment of metastatic or locally advanced disease refractory to other therapies. Pemigatinib and infigratinib are FGFR1-3 inhibitors whose mechanism of action is based on reversible binding to the FGF receptor domain, and their efficacy is limited by the development of resistance mutations in the kinase domain of the same receptor. In a multicenter, open-label, phase II study, pemigatinib showed a potential role in previously treated, locally advanced or metastatic patients; 38 (35.5% [95% CI 26.5–45.4]) patients with FGFR2 fusions or rearrangements achieved an objective response (3 complete responses and 35 partial responses) [53] with limited toxicities (hyperphosphatemia was the most common all-grade adverse event). In a phase II trial among 108 cases of pretreated iCCA patients with FGFR2 fusion or rearrangement, infigratinib showed an overall response rate (ORR) of 23.1%, with a median duration of response of 5.0 months and a median progression-free survival (PFS) of 7.3 months [54]. The most common treatment-emergent adverse events were hyperphosphatemia, eye disorders, stomatitis and fatigue.

PHOENIX-CCA2 is a multicenter phase II, open-label, single-arm study aimed at evaluating the efficacy and safety of futibatinib, an irreversible FGFR1-4 inhibitor that is less likely to induce secondary resistance mutations [55]. The trial enrolled patients with advanced or metastatic iCCA, positive for FGFR2 alterations and with disease progression after one or more lines of therapy (excluding patients previously treated with the same class of drugs). Of the 103 patients enrolled, 43 had disease response to treatment (42%; 95% CI, 32–52), including 1 patient who had a complete disease response; disease stability was observed in approximately 85 patients (83%; 95% CI, 74–89). The median duration of response was approximately 9.7 months. At a median follow-up of 17.1 months, the median PFS was 9 months (95% CI, 6.9–13.1) and the median OS was 21.7 months (95% CI, 56–75). Several phase III trials are ongoing in first-line settings for metastatic iCCA; in particular, the PROOF Trial evaluates the efficacy and safety of infigratinib versus gemcitabine and cisplatin (with a 2:1 randomization) in patients with FGFR2 gene fusions/translocations [56]. The FIGHT-302 trial, still ongoing with open recruitment, explores the role of pemigatinib versus gemcitabine plus cisplatin in first-line settings in patients with confirmed FGFR2 rearrangement [57]. TAS-120-301 (2019-004630-42 EudraCT) is a parallel two-arm, randomized study evaluating the efficacy and safety of futibatinib versus gemcitabine–cisplatin as a first-line treatment in patients with advanced, metastatic or recurrent unresectable iCCA harboring FGFR2 gene rearrangements. An Expanded Access Use of derazantinib is ongoing for patients with locally advanced, inoperable or metastatic iCCA with FGFR genomic alterations on a patient-by-patient basis, while clinical development of derazantinib is ongoing [58]. Derazantinib is an orally bioavailable multikinase inhibitor with potent pan-FGFR activity which has shown antitumor activity and a manageable safety profile in patients with advanced, unresectable iCCA with FGFR2 fusion who progressed after chemotherapy. The only registered and approved drug by the European Medicines Agency for unresectable iCCA is pemigatinib since March 2021; in April 2023, the Committee for Medicinal Products for Human Use put forward a positive opinion on the marketing authorization of futibatinib, which, nevertheless, has yet to come on the market.

Pancreatic ductal adenocarcinoma harbors FGFR aberrations in a subset of patients, ranging from approximately 4% to 5% [6], including, in a lower percentage, FGFR gene fusions and coexistent FGFR mutation and amplifications. Preclinical studies assessed anti-FGFR molecule activity against a panel of pancreatic cell lines, primary tumor explants and patient-derived xenograft models [59,60]. Dovitinib, a potent FGFR multikinase inhibitor, achieved a preclinical anticancer effect in pancreatic cancers with heightened FGFR signaling, suggesting that the efficacy may be most pronounced in cancer cells overexpressing FGFR2. In particular, the FGFR2 mRNA level, specifically the FGFR2 IIIb isoform, showed a potential predictive role in dovitinib sensitivity [52]. In the phase I/II FIGHT-101 basket study, which evaluated the safety, pharmacokinetics, pharmacodynamics and preliminary efficacy of pemigatinib as monotherapy or in combination therapy for refractory advanced malignancies, patients enrolled in part 2 (dose expansion) had measurable disease with documented FGF/FGFR alterations [61]. One in four patients with pancreatic cancer had FGFR2-USP33 fusion and maintained a response for 10.7 months. Few clinical cases showed a response to anti-FGFR therapy in patients pretreated with chemotherapy with known FGFR alterations [62,63]. In particular, a 28-year-old patient with stage IV pancreatic ductal adenocarcinoma with a confirmed FGFR2 rearrangement treated with erdafitinib, an orally administered tyrosine kinase inhibitor of FGFR1–4, had a favorable response to this treatment for more than 12 months, with an improvement of previous pancreatic-ductal-adenocarcinoma-related weight loss and a resolution of ascites and hypercalcemia [62]. The same molecule showed a 10-month complete response in a 68-year-old woman with metastatic pancreatic carcinoma harboring a FGFR2-TACC2 fusion [63].

## 4. Targeting FGFR in Gastric Cancer

FGFR2 alterations are found in different percentages in GC and esophagogastric junction adenocarcinoma (EJC), with a prevalence between 9 and 61% of patients [64,65]. In a series of 176 untreated patients with EJC, FGFR2 amplification tested using real-time PCR was present in 15% of cases, while FGFR2-positive expression by immunohistochemistry (IHC) was found in 61% of patients. Although FGFR2-positive IHC was associated with PCR amplification (*p* < 0.05), only FGFR2-positive IHC expression was associated with increased tumor depth (*p* < 0.001) and lower overall survival (OS) (*p* = 0.007) [66]. In a large cohort of GC patients receiving surgery without prior chemotherapy, FGFR2 was overexpressed in 31.1% of them. Tumors with FGFR2 positivity were more frequently associated with poorly differentiated adenocarcinoma histology, vascular invasion and more advanced stage. Moreover, FGFR2-positive patients had a higher chance of peritoneal recurrence after surgery. On the whole, FGFR2-overexpressing tumors had significantly shorter survival than nonoverexpressing cancers (*p* < 0.001 and *p* = 0.019, respectively) [67].

Different FGFR inhibitors have been used as single agents both in patients with FGFR-overexpressed and FGFR-amplified GC [68]. The pan-FGFR reversible TKI AZD4547 was tested in the phase II SHINE study. Patients were randomized 3:2 (FGFR2 gene amplification) or 1:1 (FGFR2 polysomy) to receive either the experimental oral drug or paclitaxel. AZD4547 was administered twice daily on a 2 weeks on/1 week off schedule on a 21-day cycle. A total of 71 patients were randomized and 67 received study treatment. AZD4547 did not significantly improve the primary endpoint of median PFS. Indeed, the median PFS was 1.8 months with AZD4547 and 3.5 months with paclitaxel (*p* = 0.9581) [63]. A possible explanation of these negative results came from a biomarker analysis from the study showing a significant intratumoral heterogeneity [69].

Futibatinib is a potent and irreversible inhibitor of all four FGFR isoforms (1-4). Futibatinib showed a reduced risk of developing drug resistance compared to AZD4547 and a greater inhibitory potential of incoming FGFR2 mutations causing secondary resistance [70]. Futibatinib showed activity in two different phase I studies [71]. In a phase I dose-expansion study, two out of nine patients with GC (with FGFR2 amplification and FGFR3 fusion, respectively) had a partial response to treatment, with an ORR of 22% [72]. In a different Japanese phase I study, in patients with a higher FGFR2 copy number (≥10), ORR was 36.4% and disease control rate 54.5% [73]. Bemarituzumab (FPA 144), a specific anti-FGFR2 monoclonal antibody, was tested in a phase I monotherapy study and led to an ORR of 18% (5 out of 28 patients) in patients with chemorefractory GC, FGFR2 amplification and FGFR2b overexpression [74]. In the randomized, double-blind, placebo-controlled phase II FIGHT trial, GC and EJC patients with FGFR2b overexpression (by IHC) or FGFR2 gene amplification (by ctDNA evaluation) received either bemarituzumab or placebo in combination with modified FOLFOX6 (mFOLFOX6) chemotherapy. Treatment was administered in first-line settings; HER2-positive patients were excluded and 30.2% of the prescreened patients showed FGFR2b overexpression or FGFR2 amplification and were included. At a median follow-up of 10.9 months, 155 patients were randomized. Evaluation of median PFS, the primary endpoint, was performed after 59% of the randomized patients experienced a PFS event. Treatment with bemarituzumab did not significantly improve median PFS compared to placebo (9.5 versus 7.5 months, *p* = 0.073). In a post hoc analysis with additional follow-up (12.5 months), median OS was 19.2 months for the experimental arm versus 13.5 months for placebo (hazard ratio [HR] 0.60, 95% confidence interval [CI] 0.38–0.94) [75]. Moreover, in the subset of patients with FGFR2b+ ≥10 by IHC, median OS was 25.4 months with bemarituzumab compared to 11.1 months with placebo (HR 0.41, 95% CI 0.23–0.74) [76]. The phase III FORTIDUDE-102 trial is recruiting and investigating bemarituzumab in association with mFOLFOX6 and nivolumab versus mFOLFOX6 and nivolumab in FGFR2b-overexpressed GC and EJC (NCT05111626).

## 5. Targeting FGFR in Colorectal Cancer

Approximately one-third of CRC patients carry FGFR gene alterations; for this reason, FGFR represents a promising new target for CRC treatment [77]. Some CRCs harbor FGFR genetic alterations, including copy number gains, specific mutations and mRNA overexpression. These mutations usually lead to tumor growth and resistance to TKIs [78]. FGFR2 and its isoform are highly expressed in CRC and, as a result, are correlated with tumor oncogenesis, metastasis and angiogenesis [79].

In the past few years, several selective FGFR inhibitors have been tested in clinical studies to evaluate their therapeutic effects on patients, for example AZD4547 [80], NVP-BGJ398 [81] and LY2874455 [82]. These agents competitively bind the ATP-binding pockets of FGFR in a noncovalent form, thereby preventing receptor phosphorylation and blocking signal transmission. However, these inhibitors showed selective efficacy for FGFR1-3 and do not demonstrate selectivity for FGFR4. These extremely specific bindings do not allow for their widespread administration in CRC patients. Also, some TKIs have been utilized to treat FGFR-deregulated tumors. In particular, ponatinib [83] is a novel third-generation TKI used in chronic myeloid leukemia and Philadelphia-positive acute lymphocytic leukemia; dovitinib [84] is a potent inhibitor of multiple receptor tyrosine kinases involved in tumor growth and angiogenesis, including fibroblast growth factor receptor 1 and 3, FMS-like tyrosine kinase 3, stem cell factor receptor and colony-stimulating factor receptor 1; lucitanib [85] is an oral, potent, selective inhibitor of the tyrosine kinase activity of vascular endothelial growth factor receptors 1–3, fibroblast growth factor receptors 1–3 and platelet-derived growth factor receptors alpha/beta. All these new drugs have been tested for FGFR inhibition including in CRC patients, but their further development has been restricted by their toxicity profiles (e.g., the Food and Drug Administration created a specific warning box including the risk of cardiovascular events and liver toxicity related to ponatinib). PRN1371 showed strong FGFR1-3 inhibitory activity in preclinical trials but has limited activity against common resistant gatekeeper mutants (V561M) of FGFR1 [86]. FINN-2 and FINN-3 irreversibly inhibit FGFR and have also been confirmed to be effective against FGFR1 and FGFR2 mutations but have no similar effects during clinical testing [87]. Other novel drugs, such as BLU9931 and BLU554, demonstrated strong activity against FGFR4 by targeting Cys522 in the hinge region but resulted in a weak efficacy against FGFR1-3 [88]. Recently, Liu et al. [89] described a novel inhibitor of FGFR, F1-7, which targets the FGFR pathway in colon cancer cell lines in a dose-dependent manner, thereby causing DNA damage in cells, inhibiting cell growth and metastasis and eventually leading to cell apoptosis. F1-7 is a second-generation TKI and pan-FGFR inhibitor able to competitively bind to the ATP-binding pocket, thereby inhibiting FGFR phosphorylation and blocking the activation of the downstream signaling pathway; therefore, this inhibitor could be developed as a novel anticancer drug to treat colon cancer. Pemigatinib is an oral inhibitor of FGFR1-3 with proven efficacy in FGFR-altered cholangiocarcinoma and myeloid/lymphoid neoplasms, among others [53,90]. In a recently published phase II single-arm study [91], pemigatinib improved response rates compared to historical controls in patients with refractory FGF/FGFR-altered metastatic CRC. Eligible patients received prior chemotherapy with fluoropyrimidine, oxaliplatin, irinotecan and anti-VEGF/anti-EGFR/anti-PD-1. All patients included in the trial and treated with pemigatinib had documented FGFR1-4 mutations and/or FGF/FGFR amplifications detected by tissue- and/or blood-based molecular testing; no FGFR translocations were reported. Pemigatinib demonstrated a good safety profile but an unsatisfactory efficacy in this population. Translational studies are ongoing to investigate resistance mechanisms to pemigatinib.

## 6. Discussion

Anti-FGFR therapy showed promising results in pretreated CCAs, especially in the iCCA subtype, leading to approval of pemigatinib as single agent in the presence of FGFR2 fusion or rearrangement. To date, pemigatinib is the only anti-FGFR agent available in clinical practice. In the setting of GC and EJC, the anti-FGFR2b inhibitor bemarituzumab achieved important results in combination with chemotherapy in the first-line treatment of advanced disease and a phase III trial testing the combination of doublet chemotherapy, anti-FGFR2b and anti-PD1 is ongoing. However, response rates of anti-FGFR drugs are modest and resistance appears to emerge rapidly, limiting their efficacy and causing disease progression. The most significant and satisfying results in clinical trials testing selective FGFR inhibitors in gastrointestinal cancers reported a median progression-free survival of 7–9 months in [54,55,56]; moreover, bemarituzumab in gastric cancer did not prove superior to placebo, but still reported a prolonged PFS of 9.5 months in chemorefractory patients [76]. Much less satisfactory results have been obtained thus far in other gastrointestinal tumors [92]. These data show that targeting FGFR in these malignancies offers inhomogeneous survival rates not yet overlapping the prolonged disease control and survival achieved in other malignancies, i.e., urothelial cancers [92], but still promising and worthy of further investigation.

Some studies showed the mechanisms underlying secondary resistance to selective FGFR inhibitors in FGFR-driven cancers, detecting single and multiple mutations in the FGFR tyrosine kinase domain [91]. Futibatinib, due to its mechanism of action, remains active against some of these secondary FGFR2 mutations [93]. The feedback activation of EGFR signaling plays a major role in maintaining oncogenic signaling and limiting cell death upon FGFR inhibition. Therefore, combination treatment with EGFR/ERBB inhibitors can overcome rebound activation of MEK/ERK and mTOR signaling inducing apoptosis in models that are both sensitive and resistant to single-agent treatment [92]. In patient-derived models, the synergistic combination treatment strategies were able to overcome resistance mediated by EGFR activation [93]. Given the expanding role of FGFR inhibitors in cancer care and the increasing number of studies testing combinations with chemotherapy and immune checkpoint inhibitors, it is important to consider the unique spectrum of side effects of these agents in order to prevent unnecessary dose reductions and interruptions. Indeed, anti-FGFR agents are known to cause particular toxicities related to the inhibition of the FGFR pathway that takes part in various physiological functions. The most common adverse event is hyperphosphatemia, with a prevalence of 60–76% of treated patients, related to the FGFR1 receptor that is present in proximal renal tubule cells [94]. Although the management of this condition is widely known, the correction of the phosphate imbalance very often results in a dose reduction and sometimes leads to treatment discontinuation. Moreover, fatigue, diarrhea and dermatologic and ocular toxicity are common adverse events and can lead to dose escalation or treatment interruption [95].

A current issue in the development and use of FGFR inhibitors is the validation of testing methods for FGFR2 molecular alterations. FGFR is activated in different cancers due to a wide heterogeneity of gene alterations such as fusions or rearrangements, point mutations and amplifications of the FGFR genes. In the iCCA setting, FGFR2 fusions have been reported in up to 15% of patients and are usually mutually exclusive with IDH1 mutations [96]. Reliable and accurate detection of these fusions is critical, but at present, real-world data on the performance of validated techniques are limited. Both fluorescence in situ hybridization and RNA-based next-generation sequencing are accepted tests [97]. Diversely, bemarituzumab showed the greatest efficacy in FGFR2b-overexpressed GC and EJC tested with IHC. As a result, the ongoing FORTITUDE-102 study is including patients with FGFR2b ≥ 10% 2+/3+ tumor cells as determined by centrally performed immunohistochemistry (IHC) testing, based on tumor sample.

## 7. Future Directions

There are several new anti-FGFR therapies under evaluation to overcome toxicity and resistance mutations. CPL304110 is a new tyrosine kinase inhibitor of FGFR1–3 administered orally; according to the result from a phase I clinical study presented at the ESMO Targeted Anticancer Therapies Congress 2023, the new drug showed acceptable toxicity (only 2/21 patients developed treatment-related adverse events of at least grade 3) and an encouraging response rate in heavily pretreated patients with FGFR-aberrant advanced solid malignancies [98]. In particular, the response rate was 14.3% among all patients regardless of FGFR alterations, which rose to 50.0% among patients with tumor FGFR aberrations. RLY 4008 is a potent and highly selective FGFR2 inhibitor which demonstrates a strong activity against primary and acquired FGFR2 resistance mutations in cellular assays, and potent antiproliferative effects on FGFR2-altered human tumor cell lines. In rat and dog toxicology studies, this drug was well tolerated and not associated with hyperphosphatemia or tissue mineralization at exposures significantly above those required to induce regression in all models [99]. Considering the large number of anti-FGFR molecules under study, it is important to understand how different FGFR inhibitors can best be used, in terms of sequencing or in combination with other treatments. Subgroup analyses can define the best mutational target in relation to toxicity and potentially acquired resistance to ensure the greatest benefit derived from the best patient-targeted drug, especially with the wide use and the major comprehension of the potential of next-generation sequencing and circulating tumor DNA analysis.

## 8. Conclusions

In the era of precision medicine, identifying new molecular targets for gastrointestinal tumors is mandatory. Despite the advances in cancer treatment, survival outcomes in gastrointestinal cancers remain relatively poor, partly related to the wide heterogeneity of these tumors, partly to the unsatisfying responses to novel therapeutic strategies, such as target agents and immunotherapy. Among different targets, the inhibition of the FGFR pathway is novel and one of the most promising therapeutic approaches to improve treatment outcomes in gastrointestinal tumors. Several types of FGFR-targeting agents were tested or are under investigation in clinical trials, including multikinase inhibitors, pan-FGFR inhibitors and selective FGFR inhibitors. Benefits in survival outcomes have only been established in cholangiocarcinoma thus far, but new horizons are also opening up in other gastrointestinal tumors such as gastric cancer and hepatocellular carcinoma. Preclinical and clinical data regarding iCCA are exciting and in this field, FGFR inhibition represents a practice-changing strategy; however, results regarding the use of these therapeutic options are still unsatisfactory, especially for the poor safety profile of these new drugs. Therefore, it is mandatory to investigate new agents with more manageability and less toxicity. Finally, FGFR alterations and acquired resistance mutations are frequent in gastrointestinal cancers. Therefore, more clinical trials are needed to identify new strategies to overcome resistance or cotarget alternative pathways and to restore the sensitivity of FGFR-targeting agents.

## Figures and Tables

**Figure 1 biomedicines-11-02650-f001:**
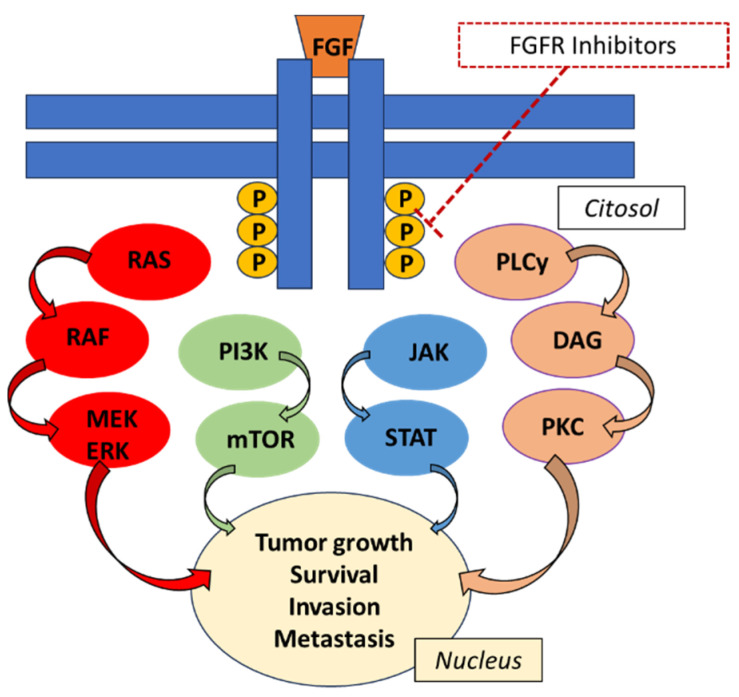
FGFR kinase inhibitors inhibit phosphorylation and the various FGFR-mediated signal transduction cascades, decreasing cell viability in cells expressing genetic alterations in FGFR, including point mutations, amplifications and fusions or rearrangements.

**Figure 2 biomedicines-11-02650-f002:**
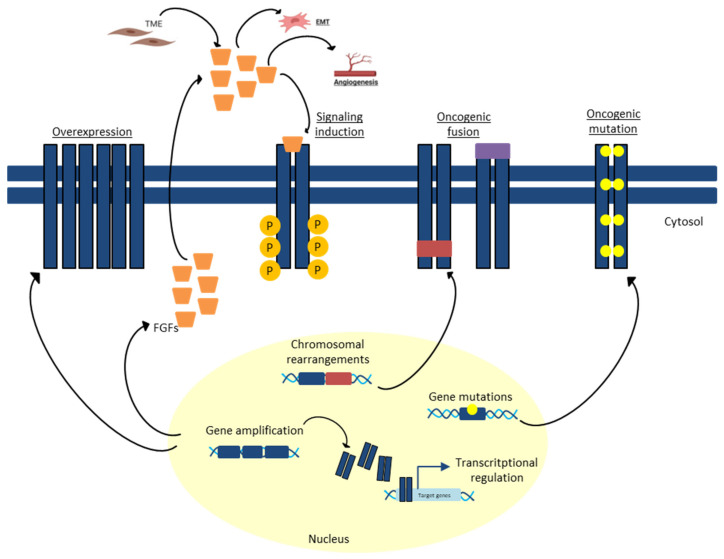
Altered mechanisms of FGFRs activation.

**Table 1 biomedicines-11-02650-t001:** Overview of the prevalence of FGFR alteration in different gastrointestinal cancers.

Alteration	Cancer Type
Mutation	0.6% GC FGFR1 mutation [6] 6% GC FGFR2 mutation [7] 57% GC FGFR4 mutation [8] 5% colorectal cancer FGFR3 mutation [7] 0.9% CCA FGFR1-4 mutation [6] 1.2% PC FGFR1-4 [6]
Fusion	20% GC FGFR3 fusion [9,10] 8% CCA FGFR2 fusion [11] 8% gastrointestinal stromal tumor FGFR1 fusion [12] 0.7% colorectal cancer FGFR2 fusion [11] 1% HCC FGFR2 fusion [11]
Copy Number Variation	1% GC FGFR1 amplification [6] 2–9% GC FGFR2 amplification [13,14] 2–3% HCC FGFR3 amplification [15] 2–3% HCC FGFR4 amplification [15] 2.6% CCA FGFR1-4 amplification [6] 2% colorectal cancer FGFR1 amplification [6] 3.5% PC FGFR1-4 amplification [6]
Altered Splicing	0.7% colon carcinoma FGFR1-4 splice variant [6] 3.5% CCA FGFR1-4 splice variant [6] 1.2% GC FGFR1-4 splice variant [6] 0.6% PC FGFR1-4 splice variant [6]

CCA = cholangiocarcinoma; FGFR = fibroblast growth factor; GC = gastric cancer; HCC = hepatocellular carcinoma; PC = pancreatic cancer.

## Data Availability

The data that support the findings of this study are available from the corresponding author upon reasonable request.

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
