# Peer review of "Targeting FGFR Pathways in Gastrointestinal Cancers: New Frontiers of Treatment"

_biomedicines, 2023, doi:10.3390/biomedicines11102650_

Round 1

Reviewer 1 Report

This review contains up-to-date information and reflects the state of the art in the field.

Minor comments:

1. The number of non-standard abbreviations is way too high. This seriously undermines the readability of the review.

2. The use of multiple indentations on e.g. page 2 but also on other pages is  aberrant. Are these all separate paragraphs? The text should be coherent and not a collection of separate thoughts or ideas.

3. The pertinent question is whether the effect of these drugs in clinical trials are clinically meaningful. 

4. PFS (progression-free survival) is unexplained in the manuscript.

Author Response

Point by point response to reviewer 1:

We would like to thank the reviewer for his meaningful and very helpful comments.

  1. The number of non-standard abbreviations is way too high. This seriously undermines the readability of the review.

Modified according to the comment.

  1. The use of multiple indentations on e.g. page 2 but also on other pages is  aberrant. Are these all separate paragraphs? The text should be coherent and not a collection of separate thoughts or ideas.

Modified according to the comment, see in particular paragraph “FGFR pathway and its alterations in gastrointestinal cancers”

  1. The pertinent question is whether the effect of these drugs in clinical trials are clinically meaningful.

This is an excellent comment so we add a new part; see in particular paragraph 6 “Discussion” and paragraph 8 “Conclusions”

  1. PFS (progression-free survival) is unexplained in the manuscript.

Modified according to the comment

Reviewer 2 Report

Manuscript entitled "Targeting FGFR pathways in gastrointestinal cancers: new frontiers of treatment"

Major issues:

1. The authors should include the description of development and clinical trials of targeted therapy for FGFRs.

2. The descriptions and a table should be included to describe the prevelence of FGFRs alteration (mutation, fusion, CNV) in different gastrointestinal cancers.

3. A figure should be included to demonstrate the altered activation mechanism of FGFRs. 

acceptable

Author Response

Point by point response to reviewer 2:

We would like to thank the reviewer for his meaningful and very helpful comments.

  1. The authors should include the description of development and clinical trials of targeted therapy for FGFRs.

We agree with this comment and we add some new parts to the review, see in particular paragraph 6 “Discussion” and paragraph 8 “Conclusions”

  1. The descriptions and a table should be included to describe the prevelence of FGFRs alteration (mutation, fusion, CNV) in different gastrointestinal cancers.

We agree with this comment and we added a table, as suggested.

  1. A figure should be included to demonstrate the altered activation mechanism of FGFRs

We add a figure showing the altered activation mechanism of FGRSs, as suggested.

Round 2

Reviewer 2 Report

The revision is improved and acceptable for publication.

Acceptable